# Diplopia Is Frequent and Associated with Motor and Non-Motor Severity in Parkinson’s Disease: Results from the COPPADIS Cohort at 2-Year Follow-Up

**DOI:** 10.3390/diagnostics11122380

**Published:** 2021-12-17

**Authors:** Diego Santos García, Lucía Naya Ríos, Teresa de Deus Fonticoba, Carlos Cores Bartolomé, Lucía García Roca, Maria Feal Painceiras, Cristina Martínez Miró, Hector Canfield, Silvia Jesús, Miquel Aguilar, Pau Pastor, Marina Cosgaya, Juan García Caldentey, Nuria Caballol, Inés Legarda, Jorge Hernández Vara, Iria Cabo, Lydia López Manzanares, Isabel González Aramburu, María A. Ávila Rivera, Víctor Gómez Mayordomo, Víctor Nogueira, Víctor Puente, Julio Dotor, Carmen Borrué, Berta Solano Vila, María Álvarez Sauco, Lydia Vela, Sonia Escalante, Esther Cubo, Francisco Carrillo Padilla, Juan C. Martínez Castrillo, Pilar Sánchez Alonso, Maria G. Alonso Losada, Nuria López Ariztegui, Itziar Gastón, Jaime Kulisevsky, Marta Blázquez Estrada, Manuel Seijo, Javier Rúiz Martínez, Caridad Valero, Mónica Kurtis, Oriol de Fábregues, Jessica González Ardura, Ruben Alonso Redondo, Carlos Ordás, Luis M. López Díaz, Darrian McAfee, Pablo Martinez-Martin, Pablo Mir

**Affiliations:** 1CHUAC, Complejo Hospitalario Universitario de A Coruña, 15006 A Coruña, Spain; lucia.naya.rios@gmail.com (L.N.R.); Carlos.Cores.Bartolome@sergas.es (C.C.B.); lucia.garcia.roca@gmail.com (L.G.R.); mjfealpainceiras@gmail.com (M.F.P.); Cristina.Martinez.Miro@sergas.es (C.M.M.); canfield.hector@gmail.com (H.C.); 2CHUF, Complejo Hospitalario Universitario de Ferrol, 15006 A Coruña, Spain; terecorreomovil@gmail.com; 3Unidad de Trastornos del Movimiento, Servicio de Neurología y Neurofisiología Clínica, Instituto de Biomedicina de Sevilla, Hospital Universitario Virgen del Rocío/CSIC/Universidad de Sevilla, 41013 Seville, Spain; smaestre-ibis@us.es (S.J.); pmir@us.es (P.M.); 4CIBERNED (Centro de Investigación Biomédica en Red Enfermedades Neurodegenerativas), 28031 Madrid, Spain; isagaramburu@gmail.com (I.G.A.); jkulisevsky@santpau.cat (J.K.); pmm650@hotmail.com (P.M.-M.); 5Hospital Universitari Mutua de Terrassa, 08221 Terrassa, Barcelona, Spain; miquelaguilar@gmail.com (M.A.); paupastor@mutuaterrassa.es (P.P.); 6Hospital Clínic de Barcelona, 08036 Barcelona, Spain; marinacosgaya@gmail.com; 7Centro Neurológico Oms 42, 07003 Palma de Mallorca, Spain; juangcaldentey@hotmail.com; 8Consorci Sanitari Integral, Hospital Moisés Broggi, 08970 Sant Joan Despí, Barcelona, Spain; nuriacaballol@hotmail.com; 9Hospital Universitario Son Espases, 07120 Palma de Mallorca, Spain; ines.legarda@ssib.es; 10Hospital Universitario Vall d’Hebron, 08035 Barcelona, Spain; hernandezvarajorge76@gmail.com (J.H.V.); odefabregues@gmail.com (O.d.F.); 11Complejo Hospitalario Universitario de Pontevedra (CHOP), 36071 Pontevedra, Spain; icabol@yahoo.es (I.C.); manuel.seijo.martinez@sergas.es (M.S.); 12Hospital Universitario La Princesa, 28006 Madrid, Spain; Lydia.veladesojo@salud.madrid.org; 13Hospital Universitario Marqués de Valdecilla, 39008 Santander, Spain; 14Consorci Sanitari Integral, Hospital General de L’Hospitalet, L’Hospitalet de Llobregat, 08906 Barcelona, Spain; asuncion.avila@sanitatintegral.org; 15Hospital Universitario Clínico San Carlos, 28040 Madrid, Spain; vicmayordomo@gmail.com; 16Hospital Da Costa, Burela, 27880 Lugo, Spain; victor.nogueira.fernandez@sergas.es; 17Hospital del Mar, 08003 Barcelona, Spain; Vpuente@parcdesalutmar.cat; 18Hospital Universitario Virgen Macarena, 41009 Sevilla, Spain; juliodotor@gmail.com; 19Hospital Infanta Sofía, 28703 Madrid, Spain; carmenborrue@hotmail.com; 20Institut d’Assistència Sanitària (IAS)—Institut Català de la Salut, 17190 Girona, Spain; berta_solano@hotmail.com; 21Hospital General Universitario de Elche, 03203 Elche, Spain; mariaalsa@hotmail.com; 22Fundación Hospital de Alcorcón, 28922 Madrid, Spain; lvela@fhalcorcon.es; 23Hospital de Tortosa Verge de la Cinta (HTVC), Tortosa, 43500 Tarragona, Spain; sescalant@yahoo.es; 24Complejo Asistencial Universitario de Burgos, 09006 Burgos, Spain; esthercubo@gmail.com; 25Hospital Universitario de Canarias, 38320 Santa Cruz de Tenerife, Spain; fcarpad@gobiernodecanarias.org; 26Hospital Universitario Ramón y Cajal, IRYCIS, 28034 Madrid, Spain; jcmcastrillo@gmail.com; 27Hospital Universitario Puerta de Hierro, 28222 Madrid, Spain; PISANCHEZAL@GMAIL.COM; 28Hospital Álvaro Cunqueiro, Complejo Hospitalario Universitario de Vigo (CHUVI), 36213 Vigo, Spain; gemavarita@gmail.com; 29Complejo Hospitalario de Toledo, 45004 Toledo, Spain; nlariztegui@gmail.com; 30Complejo Hospitalario de Navarra, 31008 Pamplona, Spain; itziar.gaston.zubimendi@cfnavarra.es; 31Hospital de Sant Pau, 08041 Barcelona, Spain; 32Hospital Universitario Central de Asturias, 33011 Oviedo, Spain; marta.blazquez.estrada@gmail.com; 33Hospital Universitario Donostia, 20014 San Sebastián, Spain; JAVIER.RUIZMARTINEZ@osakidetza.eus; 34Hospital Arnau de Vilanova, 46015 Valencia, Spain; carivalero@icloud.com; 35Hospital Ruber Internacional, 28034 Madrid, Spain; mkurtis@ruberinternacional.es; 36Hospital de Cabueñes, 33394 Gijón, Spain; jessardura@yahoo.es; 37Universitario Lucus Augusti (HULA), 27003 Lugo, Spain; alonso_1408@hotmail.es; 38Hospital Rey Juan Carlos, 28933 Madrid, Spain; carlos.ordas@quironsalud.es; 39Complejo Hospitalario Universitario de Orense (CHUO), 32005 Orense, Spain; Luis.Manuel.Lopez.Diaz@sergas.es; 40University of Maryland School of Medicine, Baltimore, MD 21201, USA; mcafeed@sas.upenn.edu

**Keywords:** changes, motor, Parkinson’s disease, phenotype, PIGD, Tremor

## Abstract

Background and objective: Diplopia is relatively common in Parkinson’s disease (PD) but is still understudied. Our aim was to analyze the frequency of diplopia in PD patients from a multicenter Spanish cohort, to compare the frequency with a control group, and to identify factors associated with it. Patients and Methods: PD patients who were recruited from January 2016 to November 2017 (baseline visit; V0) and evaluated again at a 2-year ± 30 days follow-up (V2) from 35 centers of Spain from the COPPADIS cohort were included in this longitudinal prospective study. The patients and controls were classified as “with diplopia” or “without diplopia” according to item 15 of the Non-Motor Symptoms Scale (NMSS) at V0, V1 (1-year ± 15 days), and V2 for the patients and at V0 and V2 for the controls. Results: The frequency of diplopia in the PD patients was 13.6% (94/691) at V0 (1.9% in controls [4/206]; *p* < 0.0001), 14.2% (86/604) at V1, and 17.1% (86/502) at V2 (0.8% in controls [1/124]; *p* < 0.0001), with a period prevalence of 24.9% (120/481). Visual hallucinations at any visit from V0 to V2 (OR = 2.264; 95%CI, 1.269–4.039; *p* = 0.006), a higher score on the NMSS at V0 (OR = 1.009; 95%CI, 1.012–1.024; *p* = 0.015), and a greater increase from V0 to V2 on the Unified Parkinson’s Disease Rating Scale–III (OR = 1.039; 95%CI, 1.023–1.083; *p* < 0.0001) and Neuropsychiatric Inventory (OR = 1.028; 95%CI, 1.001–1.057; *p* = 0.049) scores were independent factors associated with diplopia (R^2^ = 0.25; Hosmer and Lemeshow test, *p* = 0.716). Conclusions: Diplopia represents a frequent symptom in PD patients and is associated with motor and non-motor severity.

## 1. Introduction

Parkinson’s disease (PD) is a complex disorder with a wide variety of symptoms that have a negative impact on a patient’s quality of life (QoL) and a patient’s independence for activities of daily living (ADL) [1]. Visual impairment is reported by some patients with PD, with double vision as one of the most common complaints. Diplopia is relatively common in PD but is still understudied [2]. Its prevalence in PD ranges from 10% to 30% and is usually limited to specific situations, such as reading and looking around [2,3,4,5,6,7,8]. A recent review concluded that diplopia in PD patients is usually intermittent and binocular, and the review also identified an older age, a longer disease duration, a greater disease severity, a cognitive decline, a presence of visual hallucinations, and a higher levodopa equivalent daily dose (LEDD) as risk factors contributing to it [4]. Moreover, diplopia could have a significant impact on QoL [5]. Possible comorbidities of diplopia include myasthenia gravis and vascular disease [7]. Very recently, Hamedani et al. found, using data from 26,790 PD patients and 9257 controls in the Fox Insight Study, that 28.2% of all the PD patients reported diplopia at least once during the study (period prevalence) and identified to be older, to be non-white, to have a longer disease duration, and to have greater motor, non-motor, and daily activity limitations as factors associated with diplopia. Despite diplopia being frequent in PD, it is under-recognized [8]. On the other hand, since it is easy to identify by asking the patient, diplopia could be considered as a simple clinical marker of disease severity. 

Our aim was to analyze the frequency of diplopia in PD patients from a multicenter Spanish cohort, to compare the frequency with a control group, and to identify factors associated with it.

## 2. Methods

PD patients who were recruited from January 2016 to November 2017 (baseline visit; V0) and evaluated again at a 2-year ± 30 days follow-up (V2) from 35 centers of Spain from the COPPADIS cohort [9] were included in this study. Methodology regarding COPPADIS-2015 study can be consulted at https://bmcneurol.biomedcentral.com/articles/10.1186/s12883-016-0548-9 (accessed on 15 December 2021) [10]. This is a longitudinal-prospective, 5-year follow-up study designed to analyze natural progression of PD in which patients diagnosed with PD according to UK PD Brain Bank criteria without dementia were included [10]. 

Information on sociodemographic aspects, factors related to PD, comorbidity, and treatment was collected. Motor status, non-motor symptoms (NMS), QoL, and disability were assessed at V0 and at V2 using different validated scales: Hoenh & Yahr (H&Y); UPDRS-III and UPDRS-IV; Freezing of Gait Questionnaire [FOGQ]); Parkinson’s Disease Cognitive Rating Scale (PD-CRS); Non-Motor Symptoms Scale (NMSS); Beck Depression Inventory-II (BDI-II); Parkinson’s Disease Sleep Scale (PDSS); Neuropsychiatric Inventory (NPI); Questionnaire for Impulsive-Compulsive Disorders in Parkinson’s Disease-Rating Scale (QUIP-RS); Visual Analog Scale-Pain (VAS-Pain); Visual Analog Fatigue Scale (VAFS]); the 39-item Parkinson’s disease Questionnaire (PDQ-39); PQ-10; the EUROHIS-QOL 8-item index (EUROHIS-QOL8); ADLS (Schwab & England Activities of Daily Living Scale). In patients with motor fluctuations, the motor assessment was made during the OFF state (without medication in the last 12 h) and during the ON state. The assessment was only performed without medication in patients without motor fluctuations. The same evaluation as for the patients, except for the motor assessment, was performed in control subjects at V0 and at V2 (2 years ± 1 month). Furthermore, motor (H&Y, UPDRS-III, UPDRS-IV) and non-motor assessment (NMSS and ADLS) was conducted in PD patients at 1 year ± 1 month (V1) [10]. LEED was calculated based on the literature [11].

### 2.1. Diplopia Definition

Patients were classified as “with diplopia” or “without diplopia” according to item 15 of the NMSS [12] at V0, V1, and V2. In the control group, it was done based on the evaluation at V0 and at V2. This item is one of the 30 items of this scale and is included in domain 4 (Perceptual problems/hallucinations; items 13, 14, and 15). Symptoms were assessed over the last month. Each symptom scored with respect to severity (0 = None; 1 = Mild, symptoms present but causes little distress or disturbance to the patient; 2 = Moderate, some distress or disturbance to the patient; 3 = Severe, major source of distress or disturbance to the patient) and frequency (1 = Rarely, <1/week; 2 = Often, 1/week; 3 = Frequent, several times per week; 4 = Very frequent, daily or all the time). For each item, the score is calculated as frequency x severity, being the range from 0 (without the symptom) to 12 (the most frequent and severe). Specifically, the question of item 15 is: “Does the patient experienced double vision? (2 separate real objects and not blurred vision)”. Patients with a NMSS-item 15 score = 0 at V0, V1, and V2 were considered as “without diplopia”, whereas patients with a NMSS-item 15 score ≥ 1 (from 1 to 12) in at least one of the three visits were considered as “with diplopia”. The same was applied in the controls but only for visits V0 and V2. Diplopia burden was also calculated in PD patients. The score at V0, V1, and V2 and the sum of the score from the three visits (NMSS-Diplopia_V0+V1+V2_, from 0 to 36) was calculated. Patients reporting diplopia in the three visits were defined as patients with “persisting diplopia”. 

Based on previously published studies [5,6], the relationship between diplopia and visual hallucinations was explored. Patients were classified as “with visual hallucinations” or “without visual hallucinations” according to the item 13 of the NMSS [12]: “Does the patient indicate that he/she sees things that are not there?”. The same criteria as previously explained were considered for defining visual hallucinations. 

### 2.2. Serum Biomarkers Determination

Serum biomarkers (SB) collected from the COPPADIS cohort were analyzed for knowing if there was a relationship between any of them and the presence of diplopia. Blood sample collection for the determination of different SB included S-100b protein, tumor necrosis factor (TNF)-α, interleukin (IL)-1, IL-2, IL-6, vitamin B12, methylmalonic acid, homocysteine, uric acid, ultrasensitive CRP (US-CRP), ferritin, and iron. SB levels were determined from frozen blood samples obtained from subjects participating in the COPPADIS-2015 study from 9 centers of Spain. The extraction of the sample was carried out no longer than 3 months after the first clinical assessment (V0) in the absence of infections and/or fever. The analysis was conducted at a common laboratory: REFERENCE LABORATORY (www.reference-laboratory.es, accessed on 15 December 2021). Different methods were used: visible spectrophotometry (iron); immunoluminescence (S-100b protein, ferritin, vitamin B12, and homocysteine); enzimoimmunoassay (IL-1, IL-2, and TNF-α); immunoassay (US-CRP); mass spectrometry (methylmalonic acid); enzymatic technique (uric acid). Outliers were excluded from the analysis.

### 2.3. Data Analysis

Data were processed using SPSS 20.0 for Windows. For comparisons between groups, the Student’s *t*-test, Mann–Whitney–Wilcoxon test, chi-square test, or Fisher test were used as appropriate (distribution for variables was verified by one-sample Kolmogorov–Smirnov test). Since multiple analysis on the same dependent variable (diplopia) were conducted, Bonferroni correction was applied to reduce the instance of a false positive. Binary regression models were used for determining independent factors associated with diplopia (diplopia as dependent variable). Any variables with univariate associations with *p*-values < 0.20 were included in a multivariable model, and a backwards selection process was used to remove variables individually until all remaining variables were significant at the 0.10 level. A *p*-value < 0.05 was considered significant. 

### 2.4. Standard Protocol Approvals, Registrations, and Patient Consents

Approval from the Comité de Ética de la Investigación Clínica de Galicia from Spain (2014/534; 2 December 2014) was obtained. A written informed consent from all participants was signed. COPPADIS-2015 was classified by the AEMPS (Agencia Española del Medicamento y Productos Sanitarios) as a Post-authorization Prospective Follow-up study with the code COH-PAK-2014-01.

## 3. Results

At the baseline, 691 PD patients (62.59 ± 8.92 years old; 60.2% males) and 206 controls (60.98 ± 8.34 years old; 50% males) were considered valid for the analysis. The frequency of diplopia in PD patients was: 13.6% (94/691) at V0; 14.2% (86/604) at V1; 17.1% (86/502) at V2 (Figure 1A). At V0 and at V2, diplopia was significantly less frequent (*p* < 0.0001) in the controls than in patients (1.9% at V0 and 0.8% at V2 in the controls). In the group (N = 481; 62.62 ± 8.54 years old, from 35 to 75; 59.2% males) with assessments carried out in all the visits (V0, V1, and V2), 24.9% (120/481) of the patients reported diplopia at least once during the study (period prevalence); specifically, 11.8% (57/481) in only one visit, 6% (29/481) in two out of the three visits, and 7.1% (34/481) in all three visits (i.e., persisting diplopia) (Figure 1B). Regarding the diplopia burden in PD patients, as expected, the NMSS-Diplopia_V0+V1+V2_ score was higher in patients with persisting diplopia: diplopia in one visit, 2.64 ± 2.83 (N = 57); diplopia in two out of the three visits, 7.03 ± 6.51 (N = 29); persisting diplopia, 13.52 ± 6.76 (N = 34) (*p* < 0.0001).

With regard to visual hallucinations, they were more frequent in PD patients with diplopia than in those without diplopia in all the visits: at V0, 45.7% vs. 9.4% (*p* < 0.0001); at V1, 51.2% vs. 11% (*p* < 0.0001); at V2, 53.5% vs. 13% (*p* < 0.0001) (Figure 2A). The patients with persisting diplopia presented visual hallucinations (at least once from V0 to V2) in up to 85.3% of the cases compared to 20.8% of those without diplopia in any visit (*p* < 0.0001) (Figure 2B). A moderate correlation was observed between the NMSS-Diplopia_V0+V1+V2_ score and the NMSS-Visual hallucinations_V0+V1+V2_ score (r = 0.503; *p* < 0.0001). 

At V0, patients with diplopia (N = 120), when compared to those without diplopia (N = 361), had a more severe disease in general with worse motor (UPDRS-IV, FOGQ) and non-motor status (PD-CRS, NMSS, PDSS, VAS-PAIN, VASF-Physical, VASF-Mental) and QoL (PDQ-39SI, EUROHIS-QoL), and had a greater dependency for ADL (ADLS) (Table 1). However, there were no differences between both groups regarding age, gender, and time from symptoms onset. With regard to cognition, the PD-CRS total score and the score on the posterior-cortical sub-domain were significantly lower at the baseline in the PD patients with diplopia compared to those without diplopia (Table 2). There were also no differences in relation to the molecular markers analyzed between both groups (data not shown due to lack of interest). When changes in the symptoms from the baseline visit to the end of the follow-up were analyzed, it was observed that, in the patients with diplopia compared to those without diplopia, the mean score on the UPDRS-III and NPI increased from V0 to V2 in +6.06 ± 11.31 vs. +2.23 ± 9.41 (*p* = 0.007) and in +2.12 ± 12.35 vs. −0.03 ± 7.22 (*p* = 0.44), respectively, although it was not significant after the Bonferroni correction. 

Visual hallucinations at any visit from V0 to V2 (OR = 2.264; 95%CI, 1.269–4.039; *p* = 0.006), a higher score on the NMSS at V0 (OR = 1.009; 95%CI, 1.012–1.024; *p* = 0.015), and a greater increase from V0 to V2 in the UPDRS-III (OR = 1.039; 95%CI, 1.023–1.083; *p* < 0.0001) and NPI (OR = 1.028; 95%CI, 1.001–1.057; *p* = 0.049) scores were independent factors associated with diplopia (R^2^ = 0.25; Hosmer and Lemeshow test, *p* = 0.716) (Table 3). In the same model, when the variables were included individually as dichotomous variables, the next results were obtained (all models significant): to have a severe or very severe NMS burden at V0 (NMSS > 40), OR = 2.107 (95%CI, 1.195–3.714; *p* = 0.010); an increase from V0 to V2 in the UPDRS-III > 10 points, OR = 2.171 (95%CI, 1.094–4.297; *p* = 0.026). A significant OR for the NPI total score increase from V0 to V2 was not observed when cut-points were defined (increase in ≥5, >5, >10, >15, >20 points).

## 4. Discussion 

The present study observes that diplopia is frequent in PD patients, clearly much more frequent than in controls, and is associated with motor and non-motor severity, including visual hallucinations, which suggests that it could be used as a simple clinical marker of the disease state. In other words, a neurologist could ask about diplopia in clinical practice, and, if the patient’s response is positive, it should be taken into account that there is a greater probability of having a more advanced disease.

The prevalence of diplopia during the period of our study, 2 years, was 24.9%. It seems to be in line with previous reports, ranging from 10% to 30% [4]. Schindlbeck et al. [3] detected binocular diplopia in 37 out of 125 PD patients (29.6%) who were screened for diplopia, visual hallucinations, problems with spatial perception, contrast sensitivity, presence of blurred vision, and history of ophthalmological comorbidities via interview. In a more recent longitudinal study, Hamedani et al. [8] reported a point prevalence diplopia of 18.1% in 26,790 PD patients and 6.3% in a control group, being present at least once in up to 28.2% of all the patients during the study (period prevalence). In our cohort, diplopia for each transversal analysis ranged from 13.6% to 17.1% in the patients and was very low in the controls. The inclusion/exclusion criteria of the study with a maximum age limit of 75 years and the absence of relevant comorbidities [10] could explain the low percentage in the controls. Other studies were designed to compare patients with diplopia previously selected versus without diplopia and controls but not to estimate the prevalence [5,6]. One important point is that diplopia was defined according to one specific question that asks for this symptom (item 18 from domain 4, “Perceptual problems/hallucinations”) from the NMSS, but a specific interview for trying to identify possible causes, differential diagnosis, and other visual symptoms was not specifically conducted. In fact, this is a post hoc analysis that was not initially considered in the COPPADIS study protocol [10]. However, as we previously reported [13], this methodology is not infrequent when scales are used (i.e., freezing of gait [14], motor fluctuations [15], dysphagia [16], etc.). Moreover, the NMSS is useful for detecting not only the presence of a symptom, such as the NMS-Quest [17], but the frequency and severity as well, being correct to separate the patients as those without the symptom (score = 0) versus those with the symptom (score ≥ 1). In the largest study to date to detect diplopia in PD, the NMSQuest was used, being present in 28.2% of all the PD patients compared to 9.1% of controls [8]. Using the NMSS with our methodology (only one visit), Martinez-Martin et al., identified 72 out of 411 (17.5%) patients as PD patients with double vision [18].

We found that diplopia was associated with motor and non-motor severity. Specifically, the patients with diplopia compared to those without diplopia presented a score on different scales that was indicative of a worse status in terms of motor complications, gait disturbances, global NMS burden, cognition, sleep, pain, fatigue, QoL, and disability. Conversely, there were no differences between both groups in age, time from symptoms onset, and total number of non-antiparkinsonian drugs when used as an indirect marker of comorbidity [19]. This finding suggests that asking about diplopia could be a simple clinical marker informing us of the possible condition of the patient, regardless of age and disease duration time, which is important in PD given the great variability in clinical presentation, from mild but long-term cases to very severe cases significantly affected from the beginning. The factors previously associated with diplopia are to be older, to be non-white, to have a longer disease duration, to have greater motor, non-motor, and daily activity limitations, to have visual hallucinations, to have cognitive decline, and to have a higher LEDD [3,8]. We found visual hallucinations as a symptom associated with diplopia in our cohort. In fact, the diplopia burden (frequency × severity) was correlated to the visual hallucinations burden. Visser et al., found visual hallucinations in 44% of 35 PD patients with diplopia compared to none of the 16 PD patients without diplopia and none of the 23 healthy controls [6]. Importantly, they analyzed the diplopia subtype, selective (i.e., diplopia of single objects) versus complete diplopia (i.e., diplopia of the entire visual field) and, although oculomotor abnormalities were equally prevalent in both subtypes, they found that only the patients with selective diplopia had visual hallucinations. This finding would be in accordance with the previously reported finding by Nebe & Ebersbach [20], who suggested that the selective diplopia of isolated single objects and persons in PD is possibly related to hallucinosis and that minor ocular disturbances seem to be a triggering factor for this peculiar type of misperception. Unfortunately, the type of diplopia was not analyzed in our cohort. 

Very recently, Naumann et al. observed, in a cross-sectional study conducted in 50 non-demented PD patients with and without intermittent diplopia and in 24 healthy controls, that those with diplopia reported NMS more frequently, including more subjective cognitive problems and apathy, compared to those without diplopia [5]. Regarding cognition, visual function (pentagon copying; number location; cube analysis), but not executive function, memory, or language, was worse in the group of patients with diplopia compared to the PD patients without diplopia and controls [5]. Using the PD-CRS in a much larger sample, we observed differences at the baseline in posterior-cortical and global cognitive functions between both groups, patients with diplopia (N = 120) compared to those without diplopia (N = 361). Furthermore, to have visual hallucinations was an independent factor in the binary model associated with diplopia (OR = 2.264), as found by Schindlbeck et al. [3] in their cohort of 125 PD patients (OR = 3.5), but cognitive impairment was not. A greater NMS burden and an impairment in motor and neuropsychiatric symptoms in the short-term were the other independent factors associated with diplopia. On the basis of all this data, we suggest that asking about diplopia could be used as a very simple screening question for trying to identify PD patients that are more affected; i.e., if the answer is positive, motor and axial symptoms, NMS, QoL, autonomy for ADL, as well as visual hallucinations and psychosis should be exhaustively checked. Moreover, it is necessary to keep in mind an interaction between cognitive decline, visual impairment, and other visual symptoms, such as diplopia and visual hallucinations [21]. Finally, although this is the first study in which molecular markers were analyzed in relation to the presence of diplopia or not, any SB (i.e., inflammation, neurodegeneration, etc.) was associated with diplopia in our cohort (data not shown). Hopefully, some objective biomarkers, such as retinal changes in PD using Optical Coherence Tomography (OCT), could be useful in the future and are starting to be used in clinical trials [22].

The present study has very important limitations. As has previously been indicated, this is a post hoc analysis in which diplopia was considered based on an answer to a simple clinical question from the NMSS, but a complete neuro-ophthalmological assessment was not conducted [23], and the diplopia type and mechanism involved (motor-fluctuation related, visual hallucination related, convergence insufficiency, decompensated heterophoria, etc.) were not analyzed. Furthermore, it was not collected if the neurologists ruled out myasthenia gravis, microvascular disease, or other causes of diplopia in their clinical practice. However, the NMSS or other questionnaires were used in other studies as well [3,18,24,25,26]. To our knowledge, this is the second largest prospective longitudinal study about diplopia prevalence and the related factors in PD patients compared with controls conducted to date, after the Hamedani study [8]. Nevertheless, in the Hamedani study, a portion of the participants were recruited from education/research and online through digital marketing, and the response to the questionnaires was carried out by themselves without the supervision of the neurologist. On the other hand, the information at the 1-year and 2-year follow-up was obtained from 87.4% and 72.6% of the patients, respectively, the percentage being 60.2% after the 2-year follow-up in the controls. This limitation has been reported in other prospective studies, with maintenance rates from 61.9% to 89.3% [27,28]. In the Fox Insight Study, 26,790 PD patients were followed longitudinally and completed a median of five questionnaires at 90-day intervals (IQR: 2–7), but the information on the patient losses to follow-up was not provided [8]. For some variables, the information was not collected in all cases. Instead of a specific tool for assessing comorbidity, like the Charlson Index or others, the total number of non-antiparkinsonian medications was used as a surrogate marker of comorbidity [19], and the role of possible comorbidities conditioning visual symptoms was not taken into account. Finally, the logistic regression model used to identify the independent factors associated with diplopia in our analysis only explains 25% of the variance, but it was either low as well or not provided in other studies [3,4,8].

In conclusion, diplopia represents a frequent symptom in PD patients and is associated with motor and non-motor severity. In clinical practice, to ask about the presence of diplopia when a PD patient is evaluated could be useful for screening the disease severity. 

## Figures and Tables

**Figure 1 diagnostics-11-02380-f001:**
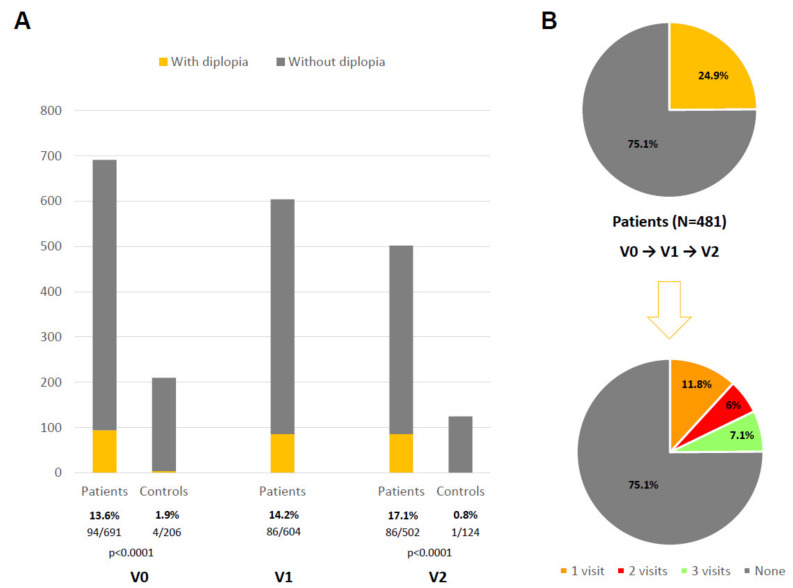
(**A**) Percentage of patients and controls reporting diplopia at different visits: V0, V1, V2. (**B**) Prevalence of diplopia during the follow-up period in all patients who completed the three visits (N = 481; V1, V2, V3) and percentage of cases presenting diplopia in only 1 visit, 2 visits, and all visits.

**Figure 2 diagnostics-11-02380-f002:**
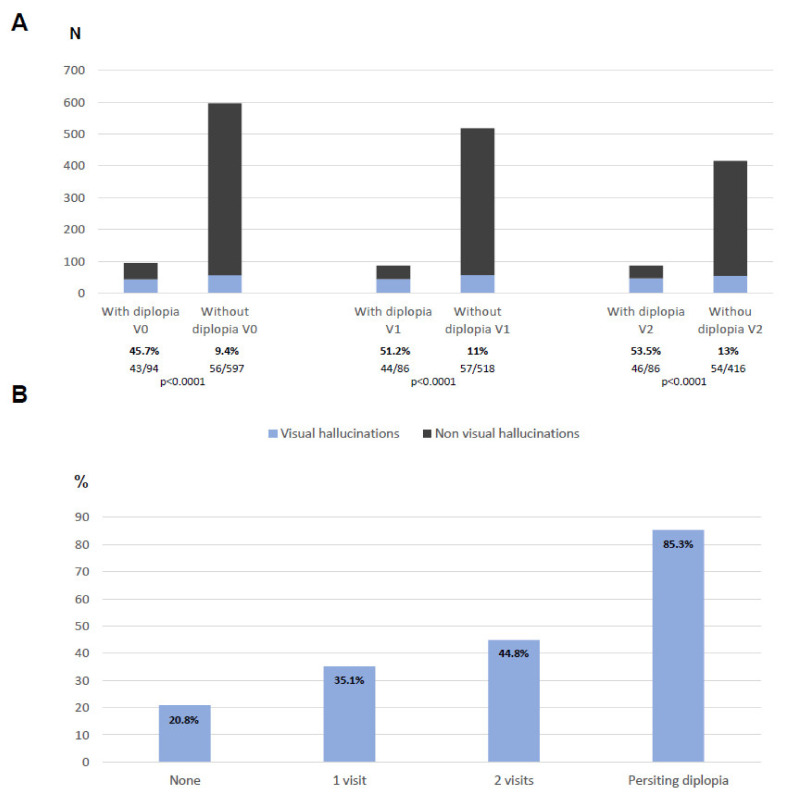
(**A**) Number of patients reporting visual hallucinations at V0, V1, and V2 when they were divided in patients with vs. without diplopia. (**B**) Percentage of patients reporting visual hallucinations at least once during the follow-up according to the times who reported diplopia.

**Table 1 diagnostics-11-02380-t001:** PD-related variables in patients with diplopia compared to those ones without diplopia (N = 481).

	All	Without Diplopia	With Diplopia	*p*
	(N = 481)	(N = 361)	(N = 120)	
**At V0 (baseline)**				
Age at baseline	62.62 ± 8.54	62.26 ± 8.78	63.69 ± 7.73	0.187
Gender (males) (%)	59	59.2	59	0.531
Time from symptoms onset	5.48 ± 4.26	5.27 ± 4.21	6.12 ± 4.35	0.312
Number of non-antipark. drugs	2.51 ± 2.34	2.46 ± 2.38	2.65 ± 2.25	0.212
Arterial hypertension (%)	32.2	29.1	41.7	0.008
Diabetes (%)	8.3	7.8	10	0.275
Dyslipemia (%)	32	31.6	33.3	0.401
Atrial fibrillation/arrhythmia (%)	4.2	3.9	5	0.38
Cardiopahy (%)	8.7	9.4	6.7	0.234
Smoking (%)	9.6	10.5	6.7	0.142
Alcohol consumption (%)	22.2	23.5	18.3	0.144
LEDD	575.3 ± 419.78	548.81 ± 408.93	653.66 ± 442.91	0.014
H&Y-OFF stage 1–2 (%)	91.1	92.2	87.8	0.112
UPDRS-III-OFF	22.37 ± 10.61	21.48 ± 10.36	24.98 ± 10.93	0.003
UPDRS-IV	2 ± 2.37	1.78 ± 2.32	2.66 ± 2.4	<0.0001
FOGQ	3.77 ± 4.6	3.24 ± 4.34	5.34 ± 5.23	<0.0001
PD-CRS	92.15 ± 15.77	93.47 ± 15.94	88.18 ± 14.61	0.001
NMSS	44.92 ± 37.87	38.14 ± 31.22	65.33 ± 47.71	<0.0001
BDI-II	8.25 ± 6.85	7.83 ± 6.58	9.52 ± 7.5	0.031
PDSS	117.27 ± 24.31	120.08 ± 21.84	108.84 ± 29.08	<0.0001
NPI	5.78 ± 7.89	5.44 ± 7.92	6.81 ± 7.74	0.023
QUIP-RS	4.35 ± 8.3	4.22 ± 8.09	4.78 ± 9.03	0.459
VAS-PAIN	2.55 ± 2.87	2.42 ± 2.86	2.95 ± 2.87	<0.0001
VASF-Physical	2.88 ± 2.7	2.64 ± 2.7	3.58 ± 2.59	<0.0001
VASF-Mental	2.11 ± 2.49	1.92 ± 2.47	2.69 ± 2.48	0.001
ADLS	88.48 ± 11.43	89.45 ± 8.78	85.58 ± 11.43	<0.0001
PDQ-39SI	16.64 ± 12.98	14.53 ± 11.59	22.97 ± 14.82	<0.0001
PQ-10	3.79 ± 0.7	3.87 ± 0.7	3.55 ± 0.64	0.054
EUROHIS-QoL	3.78 ± 0.53	3.83 ± 0.53	3.62 ± 0.52	<0.0001
**Change at V2 (V2–V0)**				
Number of non-antipark. drugs	+0.51 ± 1.51	+0.47 ± 1.47	+0.63 ± 1.61	0.333
LEDD	+194.37 ± 330.58	+190.15 ± 326.07	+206.85 ± 344.71	0.614
UPDRS-III-OFF	+3.24 ± 10.08	+2.23 ± 9.41	+6.06 ± 11.31	0.007
UPDRS-IV	+0.71 ± 2.51	+0.63 ± 2.41	+0.93 ± 2.77	0.394
FOGQ	+1.16 ± 4.2	+1.11 ± 3.96	+1.31 ± 4.88	0.347
PD-CRS	−1.6 ± 11.85	−1.17 ± 11.25	−2.87 ± 13.46	0.089
NMSS	+ 8.4 ± 34.82	+7.07 ± 25.77	+12.41 ± 53.49	0.618
BDI-II	+0.35 ± 7.84	+0.09 ± 7.12	+1.12 ± 9.66	0.487
PDSS	+0.52 ± 26.16	+0.64 ± 22.98	−0.14 ± 34.11	0.332
NPI	+0.52 ± 8.85	−0.03 ± 7.22	+2.12 ± 12.35	0.044
QUIP-RS	+0.21 ± 9.17	−0.03 ± 8.59	+1.01 ± 10.96	0.407
VAS-PAIN	+0.34 ± 3.26	+0.38 ± 3.19	+0.22 ± 3.49	0.691
VASF-Physical	+0.28 ± 3	+0.35 ± 2.85	+0.07 ± 3.44	0.499
VASF-Mental	+0.06 ± 2.81	+0.07 ± 2.77	−0.02 ± 2.95	0.644
ADLS	−4.13 ± 11.62	−3.74 ± 10.98	−5.34 ± 13.37	0.191
PDQ-39SI	+3.48 ± 12.26	+3.16 ± 10.58	+4.45 ± 16.36	0.5
PQ-10	−0.14 ± 1.73	−0.11 ± 1.76	−0.21 ± 1.67	0.385
EUROHIS-QoL	−0.01 ± 0.6	−0.01 ± 0.59	−0.03 ± 0.62	0.727

Results are expressed as mean ± SD or %. Mann–Whitney–Wilcoxon and chi-square tests were applied for assessing the relation with variables; a *p* ≤ 0.001 was considered significant (Bonferroni correction). ADLS, Schwab & England Activities of Daily Living Scale; BDI, Beck Depression Inventory-II; EUROHIS-QOL8, European Health Interview Survey-Quality of Life 8 Item-Index; FOGQ, Freezing Of Gait Questionnaire; H&Y, Hoenh & Yahr; LEED, levodopa equivalent daily dose; NMS, non-motor symptoms; NMSB, non-motor symptoms burden; NMSS, Non-Motor Symptoms Scale; NPI, Neuropsychiatric Inventory; PD, Parkinson’s disease; PD-CRS, Parkinson’s Disease Cognitive Rating Scale; PDQ-39SI, 39-item Parkinson’s Disease Quality of Life Questionnaire Summary Index; PDSS, Parkinson’s Disease Sleep Scale; QoL, Quality of life; QUIP-RS, Questionnaire for Impulsive-Compulsive Disorders in Parkinson’s Disease-Rating Scale; UPDRS, Unified Parkinson’s Disease Rating Scale; VAFS, Visual Analog Fatigue Scale; VAS-Pain, Visual Analog Scale-Pain.

**Table 2 diagnostics-11-02380-t002:** Cognitive function in patients with diplopia compared to those ones without diplopia (N = 481).

	All	Without Diplopia	With Diplopia	*p*
	(N = 481)	(N = 361)	(N = 120)	
**At V0 (baseline)**				
PD-CRS total score	92.15 ± 15.77	93.47 ± 15.94	88.18 ± 14.61	0.001
PD-CSR FS sub-score	64.32 ± 14.48	65.15 ± 14.72	61.85 ± 13.46	0.037
Immediate verbal memory	8.05 ± 2.08	8.06 ± 8.03	8.03 ± 2.05	0.784
Sustained attention	8.59 ± 14.48	8.67 ± 1.72	8.35 ± 1.95	0.063
Working memory	7.13 ± 2.27	7.15 ± 2.25	7.07 ± 2.35	0.801
Clock drawing	9.04 ± 1.61	9.09 ± 1.68	8.91 ± 1.4	0.339
Delayed verbal memory	5.47 ± 2.79	5.43 ± 2.27	5.58 ± 2.95	0.677
Alternating verbal fluency	11.38 ± 4.54	11.71 ± 4.5	10.42 ± 4.15	0.006
Action verbal fluency	14.72 ± 5.74	15.12 ± 5.69	13.51 ± 5.78	0.003
PD-CRS PC sub-score	27.83 ± 3.14	28.32 ± 2.48	26.33 ± 4.27	<0.0001
Confrontation naming	18.25 ± 2.91	18.72 ± 2.08	16.83 ± 4.29	0.005
Clock copy	9.57 ± 1.03	9.6 ± 1.03	9.5 ± 1.02	0.638
**Change at V2 (V2–V0)**				
PD-CRS total score	−1.6 ± 11.85	−1.17 ± 11.25	−2.87 ± 13.46	0.089
PD-CSR FS sub-score	−1.42 ± 10.88	−0.96 ± 10.6	−2.77 ± 11.61	0.065
Immediate verbal memory	−0.07 ± 2.45	+0.15 ± 2.58	−0.16 ± 2.02	0.306
Sustained attention	−0.43 ± 2.01	−0.39 ± 1.93	−0.66 ± 2.33	0.16
Working memory	−0.41 ± 2.09	−0.35 ± 2.13	−0.63 ± 1.93	0.292
Clock drawing	−0.2 ± 1.96	−0.12 ± 1.93	−0.57 ± 2.05	0.05
Delayed verbal memory	+0.36 ± 2.58	+0.46 ± 2.57	−0.09 ± 2.57	0.543
Alternating verbal fluency	−0.41 ± 4.08	−0.34 ± 4.18	−0.69 ± 3.62	0.09
Action verbal fluency	−0.46 ± 4.95	−0.44 ± 5.1	−0.53 ± 4.17	0.356
PD-CRS PC sub-score	−0.17 ± 2.86	−0.19 ± 2.52	−0.1 ± 3.71	0.434
Confrontation naming	+0.07 ± 2.34	−0.01 ± 2.1	+0.46 ± 3.21	0.839
Clock copy	−0.25 ± 1.48	−0.2 ± 1.51	−0.48 ± 1.38	0.076

Results are expressed as mean ± SD. Mann–Whitney–Wilcoxon and chi-square tests were applied for assessing the relation with variables; a *p* ≤ 0.002 was considered significant (Bonferroni correction). FS, fronto-subcortical; PC, posterior-cortical; PD-CRS, Parkinson’s Disease Cognitive Rating Scale.

**Table 3 diagnostics-11-02380-t003:** Binary regression model about factors associated with diplopia.

	OR ^a^	OR ^b^	95% IC ^a^	95% IC ^b^	*p* ^a^	*p* ^b^
**At V0 (baseline)**						
UPDRS-III-OFF	1.031	1.025	1.011–1.051	0.998–1.054	0.003	0.075
NMSS	1.018	1.009	1.012–1.024	1.002–1.017	<0.0001	0.015
PDSS	0.982	0.989	0.974–0.991	0.979–1.000	<0.0001	0.056
Visual hallucinations	4.076	2.264	2.628–6.324	1.269–4.039	<0.0001	0.006
**Change at V2 (V2–V0)**						
UPDRS-III	1.039	1.052	1.016–1.062	1.023–1.083	0.001	<0.0001
NPI	1.027	1.028	1.000–1.054	1.001–1.057	0.05	0.049

Dependent variable: diplopia (reporting at least once during the study). OR and 95% IC are shown. ^a^, univariate analysis; ^b^, multivariate analysis (R^2^ = 0.25; Hosmer and Lemeshow test, *p* = 0.716). NMSS, Non-motor Symptoms Scale; NPI, Neuropsychiatric Inventory; PDSS, Parkinson’s Disease Sleep Scale; UPDRS, Unified Parkinson’s Disease Rating Scale.

## Data Availability

The data that support the findings of this study are available from the corresponding author upon reasonable request. No computer coding was used in the completion of the current manuscript.

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
