# Peer review of "Diplopia Is Frequent and Associated with Motor and Non-Motor Severity in Parkinson’s Disease: Results from the COPPADIS Cohort at 2-Year Follow-Up"

_diagnostics, 2021, doi:10.3390/diagnostics11122380_

Round 1
Reviewer 1 Report
This is a very interesting study showing that diplopia is relatively frequent finding in Parkinson’s disease. Although it has been reported previously, neurologists may not be aware of this finding.
It is a bit of the a limitation that the patients did not have a neuro-ophthalmological examined and the mechanism of diplopia was not explored, however the authors address these points in the discussion.
The authors should mention that they did not exclude myasthenia gravis, microvascular disease, etc, as causes of diplopia.
Perhaps the authors overstate their findings, perhaps it is easier to ask about hallucinations directly rather than asking indirectly about diplopia, or there are more effective ways to determine severity in PD?
Author Response
Reviewer 1
This is a very interesting study showing that diplopia is relatively frequent finding in Parkinson’s disease. Although it has been reported previously, neurologists may not be aware of this finding.
It is a bit of the a limitation that the patients did not have a neuro-ophthalmological examined and the mechanism of diplopia was not explored, however the authors address these points in the discussion.
The authors should mention that they did not exclude myasthenia gravis, microvascular disease, etc, as causes of diplopia.
AUTHORS – Thank you very much for your comment. As you suggested, we clarified it and added in the text the next sentence: “Furthermore, it was not collected if the neurologists ruled out myasthenia gravis, microvascular disease or other causes of diplopia in their clinical practice”.
Perhaps the authors overstate their findings, perhaps it is easier to ask about hallucinations directly rather than asking indirectly about diplopia, or there are more effective ways to determine severity in PD?
AUTHORS – Thank you for your comment. Visual hallucinations (VH) were significantly more frequent in patients with diplopia than in those without diplopia (about 50% vs 10%). In patients with persisting diplopia the frequency was even up to 85% and to have VH was an independent factor in the binary model associated with diplopia (OR=2.264). However, we didn´t explore VH as a marker of severity disease in our cohort. Our conclusion is based on findings of this analysis and as it has been mentioned, to ask for diplopia is a simple question easy for patients to interpret. Although previous studies have reported an association between VH and longer duration and greater severity of illness (Barnes & Davis. J Neurol Neurosurg Psychiatry 2001;70:727-733), recent papers have observed that VH can be present in early PD patients and they are independently associated with dopamine agonist use, sleep quality and cognition, but not motor severity (Clegg et al. J Parkinsons Dis 2018;8:447-453; Zhu et al. J Neurol Sci 2017 Jan 15;372:471-476). Minor VH can be frequent even in the premotor phase of PD (Pagonabarraga et al. Mov Disord 2016;31:45-52) and their presence seem to be an early clinical marker of increased neurodegeneration linked to a cognitive non-motor PD phenotype (Bejr-Kasem H, et al. Eur J Neurol 2021;28:438-447). Based on our results and as a practical message, we concluded that to ask about the presence of diplopia when a PD patient is evaluated could be useful for screening disease severity.
Reviewer 2 Report
The article by Garcia DS et al titled as “Diplopia is frequent and associates with motor and non-motor severity in Parkinson ́s disease. Results from the COPPADIS cohort at 2-year follow-up” is a statistical analysis of ‘diplopia’ condition in PD patients from a multicenter Spanish cohort. Based on the COPPADIS cohort data, the authors reported a frequent diplopia condition in PD patients and also suggested the use of diplopia symptom as a clinical marker to score the disease condition/severity. The manuscript is well written, and the results have been interpreted appropriately. The present form can be accepted for publication.
Author Response
Reviewer 2
The article by Garcia DS et al titled as “Diplopia is frequent and associates with motor and non-motor severity in Parkinson ́s disease. Results from the COPPADIS cohort at 2-year follow-up” is a statistical analysis of ‘diplopia’ condition in PD patients from a multicenter Spanish cohort. Based on the COPPADIS cohort data, the authors reported a frequent diplopia condition in PD patients and also suggested the use of diplopia symptom as a clinical marker to score the disease condition/severity. The manuscript is well written, and the results have been interpreted appropriately. The present form can be accepted for publication.
AUTHORS – Thank you very much for your comment.
